# Following carpel tunnel release, what factors affect whether patients return to the same or different hand surgeon for a subsequent procedure?

**Scott J. Halperin[ID], Meera M. Dhodapkar, Neil Pathak, Peter Y. Joo, Xuan Luo, Jonathan N. Grauer[ID]***

Department of Orthopaedics and Rehabilitation, Yale School of Medicine, New Haven, CT, United States of America

* jonathan.grauer@yale.edu

## Abstract

### Background

Following carpal tunnel release (CTR), patients may be indicated for subsequent hand surgery (contralateral CTR and/or trigger finger release [TFR]). While surgeons typically take pride in patient loyalty, the rate of returning to the same hand surgeons has not been previously characterized.

### Methods

Patients undergoing CTR were isolated from 2010–2021 PearlDiver M151 dataset. Subsequent CTR or TFR were identified and characterized as being performed by the same or different surgeon, with patient factors associated with changing to a different surgeon determined by multivariable analyses.

### Results

In total, 1,121,922 CTR patients were identified. Of these, subsequent surgery was identified for 307,385 (27.4%: CTR 289,455 [94.2%] and TFR 17,930 [5.8%]). Of the patients with a subsequent surgery, 257,027 (83.6%) returned to the same surgeon and 50,358 (16.4%) changed surgeons. Multivariable analysis found factors associated with changing surgeon (in order of decreasing odds ration [OR]) to be: TFR as the second procedure (OR 2.98), time between surgeries greater than 2-years (OR 2.30), Elixhauser-Comorbidity Index (OR 1.14 per 2-point increase), and male sex (OR 1.06), with less likely hood of changing for those with Medicare (OR 0.95 relative to commercial insurance) (p<0.001 for each). Pertinent negatives included: age, Medicaid, and having a 90-day adverse event after the index procedure.

**Data Availability Statement:** "The data underlying the results presented in the current study are owned by the third party vendor PearlDiver (https://

pearldiverinc.com/). Data can be queried using Bellwether software, which is part of the PearlDiver database. The authors did not have special permission or privileges outside of those granted via payment to the vendor."

**Funding:** The authors received no specific funding for this work.

**Competing interests:** Scott J Halperin (Jane Danowski Weiss Family Foundation Fund); Meera M Dhodapkar (Richard K. Gershon, M.D. Fund at Yale University School of Medicine, Associate Editor Visual Abstracts North American Spine Society Journal); Jonathan N Grauer (North American Spine Society Journal Editor-in-Chief)

## Conclusions

Over fifteen percent of patients who required a subsequent CTR or TFR following CTR did not return to the same surgeon. Understanding what factors lead to outmigration of patients form a practice may help direct efforts for patient retention.

## Introduction

Patients evolve important relationships with their surgeons. Following carpal tunnel release (CTR), patients may be indicated for subsequent hand surgery such as contralateral CTR and/ or trigger finger release (TFR). While surgeons typically take pride in patient loyalty, the rate of returning to the same or different hand surgeons (and related patient factors) has not been previously characterized.

Many factors contribute positively to the patient-surgeon relationship. Trust is at the forefront of this relationship [1, 2] and has been associated with improved communication [3, 4], physician empathy [5], technical competency [3], and reliability [3]. In turn, this has been shown to lead to increased adherence [6, 7], recommending a physician to others [8], continued care with the same physician [8], better health outcomes [7]. While postoperative complications have been associated with poorer trust, the negative impact have been shown to be mitigated by communication [9].

The concept of patient-surgeon relationship ties into the decision of a patient returning to the same or different surgeon when further care is needed. This is quite pertinent to CTR surgery, which is a common procedure [10, 11] for which subsequent contralateral or revision CTR has been estimated to be needed by 27% of patients [12–15]. Further, there trigger finger may be predisposed for those with carpel tunnel syndrome [16–18], and TFR is another common procedure that may go on to be needed in this population [19–21].

The likelihood of patients needing subsequent hand surgery after CTR returning to the same or different hand surgeon should be an important topic for hand surgeons as a marker of patient trust and satisfaction, as well as related to maintaining patient base. This idea of patient loyalty / returning to the same surgeon has not been previously characterized in hand surgery. However, a similar analysis was performed for total joint arthroplasty, where Moore et al. found that patients with a surgery-related adverse event were more likely to switch surgeon for their subsequent total joint replacement [22].

Using a large, national, administrative database, the current study aimed to quantify and examine factors associated with whether patients return to the same surgeon or a different one for a second surgery (CTR or TFR) after their initial CTR.

## Methods

### Data source/study population

The study cohort was abstract from the M151 PearlDiver database, a large, multi-insurance, national, administrative dataset. The use of PearlDiver has been well established for orthopaedic studies [22–27], and this has been used analyzing factors that impact returning to the same or a different surgeon in other sub-areas of orthopaedics [22]. As PearlDiver outputs data in aggregated and de-identified form, our Institutional Review Board (IRB) has found studies using this database exempt from review.

The data underlying the results presented in the current study are owned by the third party vendor PealDiver (URL: https://pearldiverinc.com/). Data can be queried using Bellwether software, which is part of the PearlDiver database. The authors did not have special permission or privileges outside of those granted via payment to the vendor.

Patients undergoing CTR were identified based on Current Procedural Terminology (CPT) code 64721 (open CTR) and 29848 (endoscopic CTR). Patients were filtered for the first instance of CTR and excluded if they were less than 18 years of age or with an indication of a traumatic, infectious, or oncologic diagnosis in the 90 days prior.

Patient factors were then defined. These included age, sex, Elixhauser Comorbidity Index (ECI, a marker of overall comorbidity burden [28]), insurance plan (Commercial, Medicare, or Medicaid), and 90-day adverse event after the index procedure (as previously defined [25]). These adverse events included cardiac events, deep vein thrombosis, pulmonary embolism, sepsis, surgical site infection, acute kidney injury, pneumonia, transfusion, urinary tract infection, and wound complication.

### Subsequent surgeries

From the population undergoing index CTR, subsequent surgeries were identified. Those undergoing subsequent CTR were identified based on subsequent incident of CTR codes, and those undergoing TFR were identified based on CPT-26055. These were the two tracked procedures, as CTR is the most common hand/upper extremity procedure [29] and TFR is common and linked to CTR [29, 30]. Also, these would both be anticipated to be part of the practice of all who performed the index procedure. The time of return to surgery was aggregated into more or less than two years.

Also, physician provider was determined as being the same or different surgeon. This was defined using PearlDiver based on the provider count between the first and second surgery being one for the same surgeon or more than one for different surgeons.

### Data analysis

To gain an understanding of the factors impacting whether patients return to the same surgeon or not for their subsequent hand surgery, first a univariable analysis was performed (Students T-Test and Chi-Square where appropriate) was performed. Next, a multivariable logistic regression was used to evaluate the independent impact of each variable.

PearlDiver Bellwether was used to collect data and perform statistical analysis. Tables and figures were created using Excel. Significance was defined as p-value less than 0.05.

## Results

### Study population

A total of 1,121,922 CTR patients were identified based on the study's inclusion/exclusion criteria. Of these patients, 307,385 (27.4%) had a subsequent CTR or TFR, of which 289,455 (94.2%) had a subsequent CTR and 17,930 (5.8%) had a subsequent TFR (Fig 1). Subsequent surgeries were performed by the same surgeon for 257,027 (83.6%) and by a different surgeon for 50,358 (16.4%).

### Factors associated with changing surgeon

Relative to patients who had subsequent surgeries performed by the same surgeon, those who had subsequent surgeries performed by a different surgeon were older, sicker, of different insurance, and had greater 90-day postoperative adverse events (p<0.0001 for each).

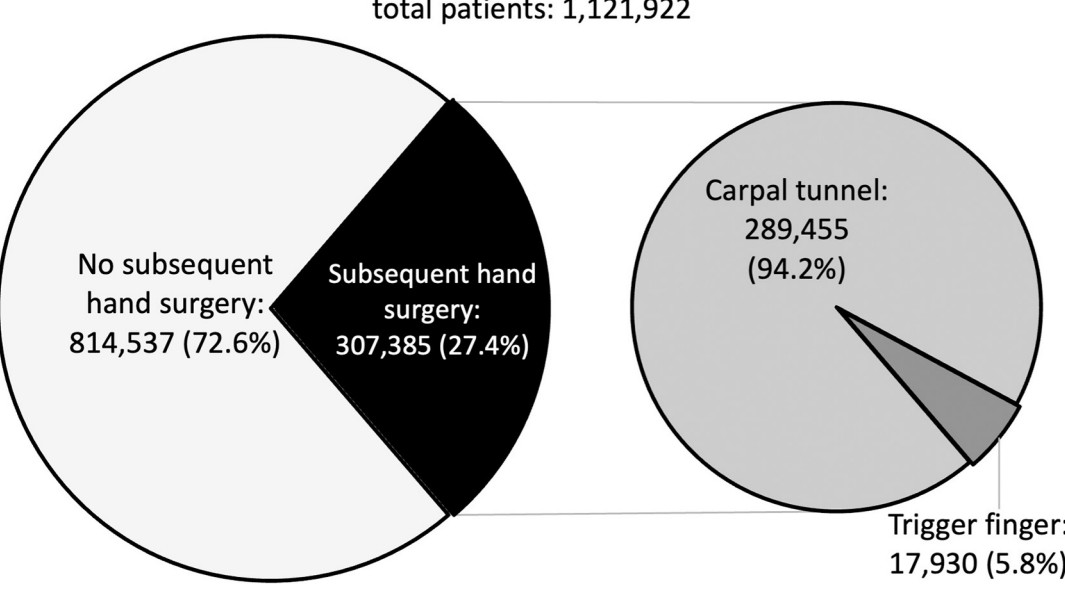

**Fig 1. This chart shows the breakdown of 1,121,922 patients after the carpal tunnel release surgery.** It shows how many had a second surgery and what type of surgery occurred.

Additionally, those going to a different surgeon were further out from index surgery (p<0.0001) and had subsequent TFR as opposed to CTR (p = 0.0002). These univariate results are shown in Table 1, left columns.

Multivariable analysis found factors associated with changing surgeon (in order of decreasing odds ration [OR]) to be: TFR as the second procedure (OR 2.98), time between surgeries greater than 2-years (OR 2.30), Elixhauser-Comorbidity Index (OR 1.14 per 2-point increase), and male sex (OR 1.06) (p<0.0001 for each). Those with Medicare insurance were less likely thon those with commercial insurance to change surgeons (OR 0.95) (p = 0.0002). Pertinent negatives included: age, Medicaid, and having a 90-day adverse event after the index procedure. These multivariate results are shown in Table 1 and Fig 2.

## Discussion

The current study evaluated the likely hood of patients returning to the same or different surgeon for a subsequent hand procedure following CTR. To our knowledge, no prior studies had assessed this tendency to return to the same or different surgeon for hand surgery and the overall numbers and associated factors should be of interest to the hand surgeon community.

Taking advantage of a large administrative database, over a million CTRs were able to be assessed. Of these 27.4% had subsequent CTR or TFR. This fits with expected numbers from the literature where the subsequent or revision CTR rate was approximately 27% [12–15]. Of those returning for subsequent surgery, 16.4% changed surgeon. This was sufficiently high to warrant further investigation to determine factors predictive of CTR patients changing surgeons for subsequent CTR or TFR (outmigration from an index surgeon's practice). Multivariable analysis was used to determine factors that were independently significant.

The largest association for changing surgeons was for returning for a TFR (compared to a CTR) with an odds ratio of 2.98. Potentially patients associate different surgeons with different procedures, but this was an unexpected finding, as these are both common procedures

**Table 1. Univariable and multivariable analysis of patients who returned to the same surgeon or a different for a subsequent hand surgery after a carpal tunnel release.**

| | Univariable | | | Multivariable | |
|---|---|---|---|---|---|
| | | | | **OR of Changing to a Different Surgeon (Compared to the Same)** | |
| | **Return to Same Surgeon: 257,027 (83.6%)** | **Return to Different Surgeon: 50,358 (16.4%)** | **P-Value** | **OR (95% CI)** | **P-Value** |
| Age (Average ± SD) | 56.4 ± 13.2 | 58.3 ± 13.1 | **<0.0001** | 1.00 (0.99–1.01) | 0.6334 |
| | | | | [per decade increase] | |
| Sex | | | 0.4597 | | |
| Female | 163,456 (63.6%) | 32,113 (63.8%) | | - | - |
| Male | 93,571 (36.4%) | 18,245 (36.2%) | | 1.06 (1.04–1.08) | **<0.0001** |
| ECI (Average ± SD) | 3.7 ± 3.1 | 4.5 ± 3.4 | **<0.0001** | 1.14 (1.14–1.15) | **<0.0001** |
| | | | | [per 2 point increase] | |
| Plan | | | **<0.0001** | | |
| Commercial | 189,866 (73.9%) | 36,767 (73.0%) | | - | - |
| Medicare | 44,050 (17.1%) | 8,947 (17.8%) | | 0.95 (0.92–0.97) | **0.0002** |
| Medicaid | 17,051 (6.6%) | 3,430 (6.8%) | | 1.02 (0.98–1.06) | 0.2853 |
| 90 Day Adverse Event After Index Procedure | 9,505 (3.7%) | 2,109 (4.2%) | **<0.0001** | 0.97 (0.93–1.02) | 0.2900 |
| Return to Surgery | | | **<0.0001** | | |
| < 2-Years | 204,639 (79.6%) | 30,829 (61.2%) | | - | - |
| > 2-Years | 52,388 (20.4%) | 19,529 (38.8%) | | 2.30 (2.25–2.35) | **<0.0001** |
| Type of Second Procedure | | | **0.0002** | | |
| Carpal Tunnel | 242,213 (94.2%) | 47,242 (93.8%) | | - | - |
| Trigger Finger | 14,814 (5.8%) | 3,116 (6.2%) | | 2.98 (2.89–3.07) | **<0.0001** |

OR: Odds Ratio

CI: Confidence Interval

SD: Standard Deviation

performed by most hand surgeons. This may be an actionable item for surgeons to make the scope of their practice better known to their patients.

The next largest association was for returning for surgery after 2-years or more with an odds ratio of 2.3. This finding seems intuitive, as time patients are more likely to have circumstance changes like moving homes, changing insurance, and more over time. Along those lines, those with Medicare insurance were less likely to change to a different surgeon (odds ration 0.95) and may be attributed to less flexibility with their insurance [31–34]. A similar trend was seen in total joint patients, where Medicare patients were less likely to change surgeons for their contralateral total hip or knee arthroplasty [22].

In terms of demographic/comorbidity factors, males were more likely to change to a different surgeon (OR 1.06). This may be due to male patients having poorer postoperative outcomes after CTR based on numbness resolution, Boston Carpal Tunnel Questionnaire score, and patient satisfaction [35]. Also, those with greater ECI were more likely to change to a different surgeon (OR 1.14 for 2-point increase). The may related to a finding from total knee arthroplasty patients, which found patients with poorer health and chronic conditions to be more likely to have lower satisfaction [36].

There were pertinent and interesting negatives for predictors of changing surgeons identified in this study. Older patients were not independently more likely to change surgeons by

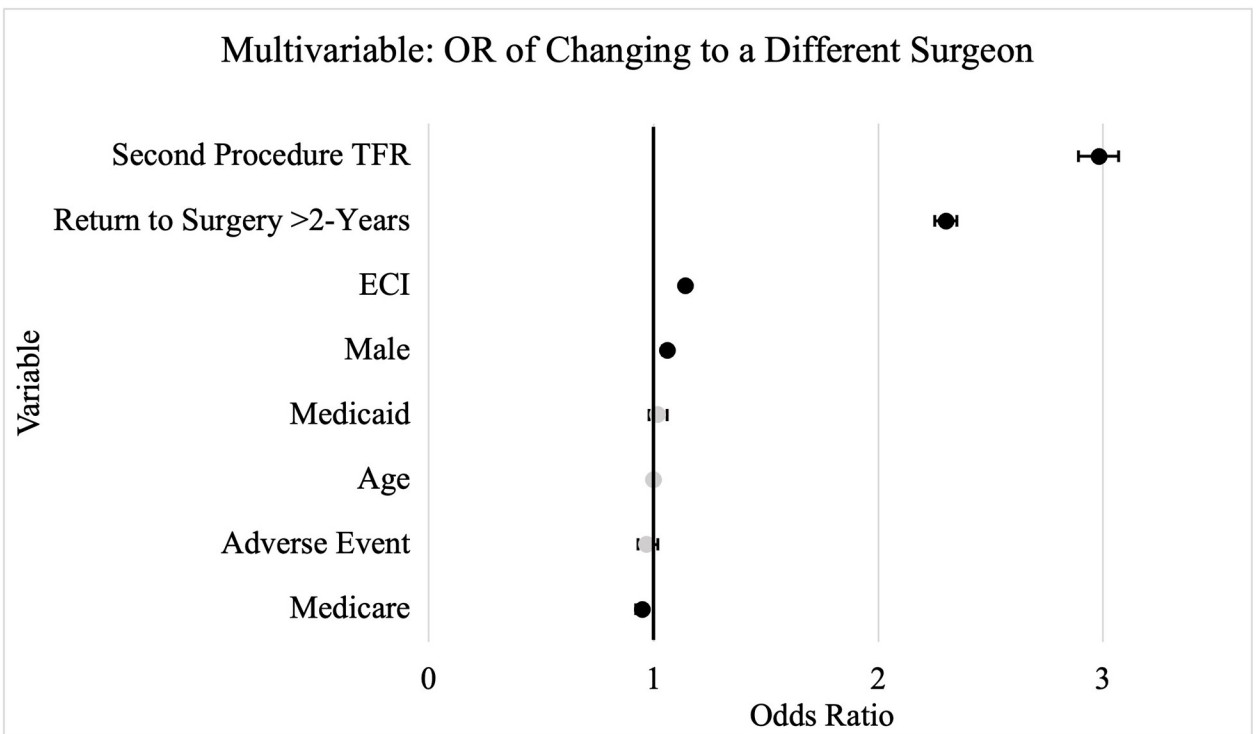

**Fig 2. This figure shows the forest plot for the multivariable logistic regression found in Table 1.** The black markers are statistically significant, and the grey ones are not.

multivariate analysis (and univariate differences were likely related to confounding insurance status). Also, those having a 90-day adverse event after the index procedure did not have a statistically significant impact on returning to a same/different surgeon. This contrasts a finding from the total joint literature [22] but may related to lesser adverse events in hand surgery [37] and strong relationships with surgeons in this subspecialty.

There are limitations to the current study. Notable is the study was retrospective in nature and dependence on administrative data. Further, the patient perceived reasons for changing physicians could not be directly assessed.

## Conclusion

In summary, over fifteen percent of patients who required a subsequent CTR or TFR following CTR did not return to the same surgeon. Factors independently associated with outmigration from an index surgeons practice were defined and may help direct efforts for patient retention.

## Supporting information

**S1 Data.**
(DOCX)

## Author Contributions

**Conceptualization:** Scott J. Halperin, Meera M. Dhodapkar, Neil Pathak, Peter Y. Joo, Xuan Luo, Jonathan N. Grauer.

**Data curation:** Scott J. Halperin, Meera M. Dhodapkar.

**Formal analysis:** Scott J. Halperin, Peter Y. Joo, Jonathan N. Grauer.

**Investigation:** Scott J. Halperin, Neil Pathak, Peter Y. Joo, Jonathan N. Grauer.

**Methodology:** Scott J. Halperin, Neil Pathak, Peter Y. Joo, Xuan Luo, Jonathan N. Grauer.

**Resources:** Jonathan N. Grauer.

**Supervision:** Xuan Luo, Jonathan N. Grauer.

**Visualization:** Jonathan N. Grauer.

**Writing – original draft:** Scott J. Halperin, Meera M. Dhodapkar.

**Writing – review & editing:** Scott J. Halperin, Meera M. Dhodapkar, Neil Pathak, Peter Y. Joo, Xuan Luo, Jonathan N. Grauer.

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
