## [Decision Letter · Decision Letter 0]

11 Dec 2023

PONE-D-23-30685Following carpel tunnel release, what factors affect whether patients return to 

the same or different hand surgeon for a subsequent procedure?PLOS ONE

Dear Dr. Grauer,

Thank you for submitting your manuscript to PLOS ONE. After careful consideration, we feel that it has merit but does not fully meet PLOS ONE’s publication criteria as it currently stands. Therefore, we invite you to submit a revised version of the manuscript that addresses the points raised during the review process.

We look forward to receiving your revised manuscript.

Kind regards,

Vojtech Kunc, Ph.D., M.D.

Academic Editor

PLOS ONE

“Scott J Halperin (Jane Danowski Weiss Family Foundation Fund); Meera M Dhodapkar (Richard K. Gershon, M.D. Fund at Yale University School of Medicine, Associate Editor Visual Abstracts North American Spine Society Journal);”

Additional Editor Comments:

Dear Authors,

I find you article in need of major revision. Please follow the recommendation of the reviewers. Also kindly mention other possible factors that could lead to patients changing surgeons from the literature (in discussion section) and add to conclusion all the factors mentioned in the results and discussion.

Reviewers' comments:

Reviewer's Responses to Questions

**Comments to the Author**

1. Is the manuscript technically sound, and do the data support the conclusions?

Reviewer #1: Partly

Reviewer #2: Yes

2. Has the statistical analysis been performed appropriately and rigorously? 

Reviewer #1: Yes

Reviewer #2: Yes

3. Have the authors made all data underlying the findings in their manuscript fully available?

Reviewer #1: No

Reviewer #2: No

4. Is the manuscript presented in an intelligible fashion and written in standard English?

Reviewer #1: Yes

Reviewer #2: Yes

5. Review Comments to the Author

Reviewer #1: The authors set out to answer a very interesting question about patient retention after simple hand surgery procedure. Finding the correct dataset for a question like this is a challenge, and I appreciate their attempt to do this with PearlDiver. I think that at the core the very simple question of "how many changed surgeon" is answerable with these data. I worry that the authors have then gone beyond the data and overreported the rest of the available results that are skewed largely due to database shortcoming. I believe some changes can be made to improve on this.

1) did you select for isolated CTR procedures (64721 or 29848 only)? Or were other CPT codes also included in the cases pulled? This analysis should be of isolated procedures (for initial and subsequent)

2) did you use any criteria for duration in pearldiver? Minimum months that patient was actually enrolled in an insurance plan and therefore included in pearldiver before and after surgery for study inclusion? This is a very important piece of the analysis especially if you are looking at multiple years duration you might have substantial patient loss unrelated to your specific research question

3) did you look at patient geography? Patients that moved to a new location? I would say those patients should be excluded

4) what were the 90 day adverse events included in the study? Defined by?

5) I assume that the medicare and Medicaid patients included in pearldiver are a unique subset of these groups. I am less familiar with inclusion details for PearlDiver but authors should clarify these details (eg all medicare advantage? Any medicare under age 65? Etc)

6)Table 1 Sex row seems to have something wrong with the numbers, they don’t really add up properly

7) did you include a minimum duration between surgeries filter? Seems unlikely someone would go to a different surgeon within a month or two. Might help the analysis of impactful variables to remove all the patients that had surgery in short time period as this likely skews results in a cohort that isnt really of interest to the primary research question.

Reviewer #2: I would like to congratulate the authors for such a nice study of the huge cohort. The study design is appropriate, and the methodology followed is acceptable. The research question is well defined. No major deficiencies were found in the manuscript. Referencing is adequate. English and grammer is adequate. Plagiarism is not checked by me.

6. PLOS authors have the option to publish the peer review history of their article (what does this mean?). If published, this will include your full peer review and any attached files.

Reviewer #1: No

Reviewer #2: **Yes: **Adil Asghar

---

## [Author Response · Author response to Decision Letter 0]

1 Oct 2024

Note: Our response document is attached in the files.

Thank you for taking the time to consider our study: “Following carpel tunnel release, what factors affect whether patients return to the same or different hand surgeon for a subsequent procedure?.” We are writing here to respond to the comments made by the reviewers after submission to PLOS ONE. We appreciate your comments and feedback on our manuscript. Below is a point-by-point response to their suggestions and comments:

Comment: Please ensure that your manuscript meets PLOS ONE's style requirements, including those for file naming. The PLOS ONE style templates can be found at https://journals.plos.org/plosone/s/file?id=wjVg/PLOSOne_formatting_sample_main_body.pdf and https://journals.plos.org/plosone/s/file?id=ba62/PLOSOne_formatting_sample_title_authors_affiliations.pdf

Response: The manuscript was edited to comply with PLOS ONE’s style requirements. 

Comment: Note from Emily Chenette, Editor in Chief of PLOS ONE, and Iain Hrynaszkiewicz, Director of Open Research Solutions at PLOS: Did you know that depositing data in a repository is associated with up to a 25% citation advantage (https://doi.org/10.1371/journal.pone.0230416)? If you’ve not already done so, consider depositing your raw data in a repository to ensure your work is read, appreciated and cited by the largest possible audience. You’ll also earn an Accessible Data icon on your published paper if you deposit your data in any participating repository (https://plos.org/open-science/open-data/#accessible-data).

Response: Thank you for this note.

Comment: Thank you for stating the following financial disclosure:

“Scott J Halperin (Jane Danowski Weiss Family Foundation Fund); Meera M Dhodapkar (Richard K. Gershon, M.D. Fund at Yale University School of Medicine, Associate Editor Visual Abstracts North American Spine Society Journal);”

Response: You are welcome. 

Comment: Please clarify the sources of funding (financial or material support) for your study. List the grants or organizations that supported your study, including funding received from your institution.

Response: No additional funding received. 

Comment: State what role the funders took in the study. If the funders had no role in your study, please state: “The funders had no role in study design, data collection and analysis, decision to publish, or preparation of the manuscript.”

Response: n/a

Comment: If any authors received a salary from any of your funders, please state which authors and which funders.

Response: n/a

Comment: If you did not receive any funding for this study, please state: “The authors received no specific funding for this work.”

Response: This has been added.

Comment: 

Response: The “raw” data is restricted for the protection of patient privacy under HIPAA. Any cohort of patients with less than 11 people is only shown as less than 11 to protect patient privacy. However, we are able to send our output files with the data collected, upon request.

Reviewer #1: 

Comment: did you select for isolated CTR procedures (64721 or 29848 only)? Or were other CPT codes also included in the cases pulled? This analysis should be of isolated procedures (for initial and subsequent)

Response: Thank you for your comment. Our cohort was selected for the first instance of both of these CPT codes. Patients did not have an indication of trauma, neoplasm, or infection prior to surgery. These were not only isolated, but we feel that our data provides insight into surgeon loyalty. 

Comment: did you use any criteria for duration in pearldiver? Minimum months that patient was actually enrolled in an insurance plan and therefore included in pearldiver before and after surgery for study inclusion? This is a very important piece of the analysis especially if you are looking at multiple years duration you might have substantial patient loss unrelated to your specific research question

Response: Thank you for your comment. We did not have a minimum months that the patient was in the dataset. 

Comment: did you look at patient geography? Patients that moved to a new location? I would say those patients should be excluded

Response: Thank you for your comment. We attempted to look at the patient’s geography, but were not able to examine the change in geography between surgeries. 

Comment: what were the 90 day adverse events included in the study? Defined by?

Response: Thank you for your comment. These were defined by a previous study (cited in the current manuscript). A further description has been added. 

Comment: I assume that the medicare and Medicaid patients included in pearldiver are a unique subset of these groups. I am less familiar with inclusion details for PearlDiver but authors should clarify these details (eg all medicare advantage? Any medicare under age 65? Etc)

Response: Thank you for this question. The database does include patients with Medicare advantage, but these will be documented as “commercial” insurance. And there are patients on Medicare under the age of 65, although this is a limited group. 

Comment: Table 1 Sex row seems to have something wrong with the numbers, they don’t really add up properly

Response: Thank you for pointing this out. This has been corrected. 

Comment: did you include a minimum duration between surgeries filter? Seems unlikely someone would go to a different surgeon within a month or two. Might help the analysis of impactful variables to remove all the patients that had surgery in short time period as this likely skews results in a cohort that isnt really of interest to the primary research question.

Response: Thank you for this question. We did not include a minimum duration between surgeries. Our thoughts were that patients could change surgeons even with a short amount of time if they were disappointed with their care or for other reasons. 

Reviewer #2: 

Comment: I would like to congratulate the authors for such a nice study of the huge cohort. The study design is appropriate, and the methodology followed is acceptable. The research question is well defined. No major deficiencies were found in the manuscript. Referencing is adequate. English and grammer is adequate. Plagiarism is not checked by me.

Response: Thank you for your review.

---

## [Editor Report · Decision Letter 1]

2 Oct 2024

Following carpel tunnel release, what factors affect whether patients return to 

the same or different hand surgeon for a subsequent procedure?

PONE-D-23-30685R1

Dear Dr. Grauer,

We’re pleased to inform you that your manuscript has been judged scientifically suitable for publication and will be formally accepted for publication once it meets all outstanding technical requirements.

Kind regards,

Vojtech Kunc, Ph.D., M.D.

Academic Editor

PLOS ONE

Additional Editor Comments (optional):

Dear Authors,

Thank you for following the comments of Reviewer 1. I find your study well written and worth publishing in PLOS ONE.

---

## [Editor Report · Acceptance letter]

10 Oct 2024

PONE-D-23-30685R1 

PLOS ONE

Dear Dr. Grauer, 

I'm pleased to inform you that your manuscript has been deemed suitable for publication in PLOS ONE. Congratulations! Your manuscript is now being handed over to our production team.

Kind regards, 

on behalf of

Dr. Vojtech Kunc 

Academic Editor

PLOS ONE